# Behavioral and Immunohistochemical Evidence for Suppressive Effects of Goshajinkigan on Salicylate-Induced Tinnitus in Rats

**DOI:** 10.3390/brainsci12050587

**Published:** 2022-04-30

**Authors:** Koichi Kitano, Akinori Yamashita, Taketoshi Sugimura, Tadao Okayasu, Masaharu Sakagami, Daisuke Osaki, Tadashi Kitahara, Yasuhiko Saito

**Affiliations:** 1Department of Otolaryngology-Head and Neck Surgery, Nara Medical University, 840 Shijo-cho, Kashihara 634-8521, Nara, Japan; k-kitano@naramed-u.ac.jp (K.K.); akinori@naramed-u.ac.jp (A.Y.); tokayasu@naramed-u.ac.jp (T.O.); m.sakagami@naramed-u.ac.jp (M.S.); dosaki@naramed-u.ac.jp (D.O.); 2Department of Neurophysiology, Nara Medical University, 840 Shijo-cho, Kashihara 634-8521, Nara, Japan; sugimura@naramed-u.ac.jp

**Keywords:** tinnitus, salicylate, animal model, Kampo, goshajinkigan, c-Fos, rat

## Abstract

Many people are affected by tinnitus, a sensation of ringing in the ear despite the absence of external sound. Goshajinkigan (GJG) is one of the formulations of Japanese traditional herbal medicine and is prescribed for the palliative treatment of patients with tinnitus. Although GJG is clinically effective in these patients, its behavioral effects and the underlying neuroanatomical substrate have not been modeled in animals. We modeled tinnitus using salicylate-treated rats, demonstrated the effectiveness of GJG on tinnitus, and examined the underlying neuronal substrate with c-Fos expression. Intraperitoneal injection of sodium salicylate (400 mg/kg) into rats for three consecutive days significantly increased false positive scores, which were used to assess tinnitus behavior. When GJG was orally administered one hour after each salicylate injection, the increase in tinnitus behavior was suppressed. The analysis of c-Fos expression in auditory-related brain areas revealed that GJG significantly reduced the salicylate-induced increase in the number of c-Fos-expressing cells in the auditory cortices, inferior colliculus, and dorsal cochlear nucleus. These results suggest a suppressive effect of GJG on salicylate-induced tinnitus in animal models.

## 1. Introduction

Many people are affected nowadays by tinnitus, a ringing sensation in the ear despite the absence of external sound. The prevalence of symptomatic tinnitus in the general population is in the range of 3.0–30.9% [1]. A population-based cohort study reported that in Japan, 11.9% of adult people experienced tinnitus [2]. Because symptoms of tinnitus are mostly subjective, its objective examination and effective evaluation of therapeutic strategies have been difficult to establish. For an objective evaluation of tinnitus, the study of its mechanisms, and development of preventive/therapeutical interventions, animal models of tinnitus can be useful. So far, tinnitus has been induced in animals by exposure to intense noise or administration of ototoxic drugs such as salicylate and quinine [3,4,5]. Tinnitus induction has been assessed using various behavioral tests, namely unconditioned gap detection startle reflex procedure [6], conditioned lick suppression [7,8], conditioned lever response suppression [9,10], and conditioned pole jumping avoidance [11].

Goshajinkigan (GJG; TJ-107, Tsumura & Co., Tokyo, Japan) is one of the popular Japanese traditional herbal medicines, the so-called Kampo medicines. GJG is composed of 10 herbal medicines in fixed proportions: Rehmanniae radix 5.0 g, Achyranthis radix 3.0 g, Corni fructus 3.0 g, Moutan cortex 3.0 g, Alismatis rhizome 3.0 g, Dioscoreae rhizome 3.0 g, Plantaginis semen 3.0 g, Hoelen 3.0 g, processed Aconiti tuber 1.0 g, and Cinnamomi cortex 1.0 g. GJG is usually prescribed for lower back pain, numbness in the extremities, dysuria, and diabetic neuropathy [12,13,14]. Traditionally, GJG has been palliatively used to treat patients for tinnitus. Ohnishi et al. [15] reported that in a multicenter clinical trial, GJG improved tinnitus symptoms in 66.7% of patients. Although GJG appears to be clinically effective for patients with tinnitus, no sufficient evidence exists in animal models. If GJG were applicable to animal models, the GJG effects on tinnitus could be objectively assessed, and its suppression of tinnitus may provide clues for the mechanisms responsible for tinnitus development and progression.

In the present study, we examined the effects of GJG on tinnitus in an animal model: footshock avoidance behavior conditioned by auditory stimuli [16] in salicylate-treated rats. Previous studies have shown that salicylate treatment increases the expression of c-Fos protein in several brain areas, including those related to auditory processing [17,18,19,20,21]. Therefore, we examined changes in c-Fos expression in auditory-related brain areas following oral GJG administration to rats that had received salicylate.

## 2. Materials and Methods

### 2.1. Animals

Experimental procedures involving animals were in accordance with the National Institutes of Health (NIH) guidelines for the Care and Use of Laboratory Animals and were approved by the Animal Care and Use Committee at Nara Medical University. All efforts were made to minimize the suffering of animals and to limit the number of animals used. A total of 150 male Wistar rats (8–23 postnatal weeks; Japan SLC, Hamamatsu, Japan) with body weights of 170–360 g was used in this study. Rats were individually housed in a temperature-controlled room at a constant 12 h light/dark cycle (lights off at 8:00 PM) and had free access to food and tap water. All behavioral tests were conducted during the animals’ activity period (dark phase) at approximately the same time of the day.

### 2.2. Behavioral Assessment of Tinnitus

To assess tinnitus in rats, we used the behavioral model of salicylate-induced tinnitus described by Kizawa et al. [16], which was a modified version of the protocol reported by Guitton et al. [11]. Rats were trained to respond to a conditioned stimulus. Conditioning and behavioral testing were performed using a shuttle box (L: 20 cm, W: 48 cm, H: 20 cm) in a soundproof room. The box consisted of two chambers separated by a 3-cm high wall. Each chamber had an electrified floor where electrical shocks could be applied independently. The paradigm was so escapable that rats easily moved to the next chamber.

A 5 s pure tone sound with a frequency of 12, 16, or 20 kHz was used as the conditioned stimulus, and a 3.7-mA electrical footshock for a maximum of 30 s was given as the unconditioned stimulus. The footshock started 1 s after the end of the 5 s sound stimulus and was applied to the room the rat stayed in when the sound stimulus was presented. The footshock was stopped when the rat moved to the opposite room to escape the footshock. One trial consisted of a sound presentation followed by a footshock. The intertrial interval (silent period) was at least 1 min. A session of 10 trials lasted 15–20 min (Figure 1A). The ratio of correct avoidance movements in response to sound stimuli within 6 s (5 s during the sound presentation plus 1 s before the electrical footshock) was designated as the active avoidance (AA) score. The number of movements to the opposite side without sound stimulation during the intertrial interval was defined as the false positive (FP) score. The more the rats experienced tinnitus, the more the FP scores increased. A lower AA score indicated the occurrence of hearing loss [16]. When AA scores reached 80% or more in three consecutive sessions, the conditioning was complete, and rats underwent behavioral tests.

The behavioral test, composed of 10 trials for 15 min, was performed once daily for 4 days, at the same time of the day (Figure 1B). Animals received daily intraperitoneal (i.p.) injections of 1 mL saline containing 400 mg/kg sodium salicylate (SS; FUJIFILM Wako Pure Chemical Industries, Ltd., Osaka, Japan) on days 1–3. Behavioral measurements were performed 2 h after injections. To check the reversibility of salicylate effects [9,16], rats underwent an additional test without salicylate injection on day 4. The AA and FP scores were determined throughout the behavioral test. When the AA scores were above 80% or more throughout the 4 days, the FP scores on day 3 were adopted as data.

We conditioned 96 rats for verification of the behavioral tests. According to Kizawa et al. [16], FP scores in salicylate-treated rats are dependent on sound intensity and frequency, and the most appropriate conditioning is at 60 dB sound pressure level (SPL) and 16 kHz. To verify the sound intensity and frequency in our experiments, we conducted the behavioral tests using the sound stimuli at intensities of 40 (*n* = 19), 60 (*n* = 31) and 80 (*n* = 12) dB SPL (16 kHz), and at frequencies of 12 (*n* = 17), 16 (previously shown), and 20 (*n* = 17) kHz (60 dB SPL) (Figure 1(B1)).

Moreover, we examined the behavioral tests following GJG administration. For the control group (*n* = 9), rats received daily i.p. saline injections following oral administration of DW. Salicylate-treated rats were divided into three groups: SS + DW group (*n* = 9), SS + GJG (0.3) group (*n* = 9), and SS + GJG (1.0) group (*n* = 9). In addition to i.p. salicylate injections, rats received daily oral administration of distilled water (DW, 10 mL/kg) with or without GJG (0.3 or 1.0 g/kg) on days 1–3 (Figure 1(B2)). The GJG dose was determined according to the results of previous studies [22,23,24]. GJG was administered 1 h after the injection of salicylate. One hour after GJG administration, the behavioral test was started. The intensity and frequency of the sound stimuli were set at 60 dB SPL and 16 kHz, respectively, as described previously [16].

### 2.3. Immunohistochemical Analysis of c-Fos Expression in the Auditory Pathway

Separate from the behavioral tests, 18 rats were divided into three groups for the immunohistochemical analysis: (1) Saline + DW group, (2) SS + DW group, and (3) SS + GJG (1.0 g/kg) group (*n* = 6 in each group). Saline or GJG (1.0 g/kg) was orally administered to rats 1 h after the injection of salicylate on days 1–3 of Figure 1(B2). One hour after DW or GJG administration on day 3, rats were anesthetized with isoflurane inhalation and a mixture of medetomidine (0.15 mg/kg), midazolam (2.0 mg/kg), and butorphanol (2.5 mg/kg) i.p. injection, and transcardially perfused with 0.01 M phosphate-buffered saline (PBS, pH 7.4), followed by 4% paraformaldehyde in 0.1 M phosphate buffer. After 18–21 h fixation, the brains were cut into 50-µm coronal sections using a microslicer (Zero 1; Dosaka, Kyoto, Japan). Brain slices were obtained from the auditory-related brain regions, including the primary auditory cortex (A1), secondary auditory cortex dorsal and ventral zone (AuD and AuV), dorsal cochlear nucleus (DC), central nucleus of the inferior colliculus (ICc), and medial geniculate body (MGB). The approximate rostrocaudal locations of the brain regions from the bregma were determined as follows: from −5.15 mm to −4.85 mm in the A1, AuD, and AuV, from −11.04 mm to −10.89 mm in the DC, from −8.91 mm to −8.61 mm in the ICc, and from −5.15 mm to −4.85 mm in the MGB, according to the rat brain atlas [25]. Of the six slices that were usually obtained, three well-prepared slices proceeded to immunohistochemistry. In this study, we used c-Fos rabbit mAb (1:1000; Cell Signaling Technology, Danvers, MA, USA #2250) as the primary antibody and goat anti-rabbit IgG antibody (1:200; Thermo Fisher Scientific, Waltham, MA, USA A11008) as the secondary antibody. The slices were washed with 25 mM PBS, blocked in PBS containing 5% bovine serum albumin, 0.1% Triton X-100, and 0.1% NaN_3_ for 1 h, and incubated with the primary antibody overnight at room temperature. Then, they were washed in PBS and incubated with the secondary antibody for 2 h. After washing in PBS, the slices were counterstained with Red fluorescent Nissl stain (1:200, Neurotrace 530/615; Thermo Fisher Scientific, Waltham, MA, USA) for 1 h and mounted on MAS-coated slides (Matsunami Glass, Osaka, Japan) using antifade medium (ProLong Gold antifade reagent; Invitrogen, Carlsbad, CA, USA).

### 2.4. Observation and Data Analysis of Immunohistochemical Stainings

Brain sections were observed under a laser-scanning confocal microscope (C2+, Nikon). Confocal images observed with a 10× (for the AC and MGB) or 20× (for the DC and IC) objective were captured and stitched together using Nikon NIS Elements software. The numbers of c-Fos-positive nuclei were counted automatically using the “Analyze Particles” function of Fiji software [26]. In the auditory cortices, layer 4 showed a higher cell density than the surrounding layers 3 and 5, which was evident in the Nissl stainings. Therefore, layer 4 was used as a landmark, and layers 1–4 and 5–6 were determined as the superficial and deep layers, respectively.

### 2.5. Statical Analysis

All values are shown as the median [interquartile range (IQR) 25–75]. The number (*n*) refers to the number of rats analyzed unless otherwise noted. Statistical analysis was performed using the Mann–Whitney *U* test or the Kruskal–Wallis test followed by Dunn’s test for multiple comparisons, and the adjusted *p* value was calculated. Differences in variance among groups were assessed using Bartlett’s test. The Wilcoxon test was used to compare data on day 3 and day 4 (Table 1). All analyses were performed using GraphPad Prism ver.9 (GraphPad Software, San Diego, CA). The threshold for statistical significance was defined as *p* < 0.05.

## 3. Results

### 3.1. Effect of Sound Intensity and Frequency on FP

At a stimulus of 16-kHz frequency, comparison of the FP scores on day 3 among groups exposed to different sound intensities revealed that the scores were higher at 40 dB and 60 dB than at 80 dB (* *p* = 0.049 in 40 dB vs. 80 dB, *** *p* = 0.0003 in 60 dB vs. 80 dB, Dunn’s test; Figure 2A). The FP scores at 60 dB were not significantly different from those at 40 dB (*p* = 0.385, Dunn’s test). The AA score was not significantly different among the three different intensities (*p* > 0.999 in 40 dB vs. 60 dB, *p* > 0.999 in 40 dB vs. 80 dB, *p* = 0.785 in 60 dB vs. 80 dB, Dunn’s test).

At a stimulus of an intensity of 60 dB SPL, comparison of the FP scores on day 3 among groups exposed to different sound frequencies revealed that no significant difference was observed among the three frequencies tested (*p* > 0.999 in 12 kHz vs. 16 kHz, *p* = 0.083 in 12 kHz vs. 20 kHz, *p* = 0.100 in 16 kHz vs. 20 kHz, Dunn’s test; Figure 2B). The AA scores were also not significantly different among the three frequencies (*p* = 0.119 in 12 kHz vs. 16 kHz, *p* = 0.275 in 12 kHz vs. 20 kHz, *p* > 0.999 in 16 kHz vs. 20 kHz, Dunn’s test). On day 4, when salicylate was not administered to rats, the FP scores significantly decreased compared with the scores measured on the preceding day (Table 1), confirming a reversible salicylate effect in this behavioral test [9,16].

### 3.2. Effect of GJG on FP scores

Salicylate injection caused an increase in FP scores at 16 kHz and 60 dB, i.e., the number of movements to the opposite test chamber without a sound stimulus (*** *p* = 0.0002, Mann–Whitney *U* test; Figure 3A).

When a high GJG dose (1.0) was administered to salicylate-treated rats, the FP scores on day 3 in the SS + GJG (1.0) group were significantly lower than those in the SS + DW group (**p* = 0.021, Dunn’s test; Figure 3B). Administration of low-dose GJG (0.3) did not lead to significant differences in FP scores (*p* = 0.275 in SS + DW vs. SS + GJG (0.3), *p* = 0.948 in SS + GJG (0.3) vs. SS + GJG (1.0), respectively, Dunn’s test). The AA scores were higher than 80% and not significantly different among these groups (Figure 3C,D and Appendix A).

### 3.3. Effect of GJG on c-Fos Expression

To clarify whether the behavioral change in salicylate-treated rats by GJG administration was attributable to changes in neuronal activity, we investigated the expression of c-Fos in several brain areas related to auditory functions (Figure 4A). After salicylate injection, the number of c-Fos-expressing cells was considerably increased in the primary (A1) and secondary (AuD and AuV) auditory cortices (Figure 4B). The numbers of c-Fos-expressing cells in salicylate-treated rats (SS + DW group) were significantly higher than those in control rats (Saline + DW group) in the A1 (**** *p* < 0.0001, Mann–Whitney *U* test), AuD (**** *p* < 0.0001, Mann–Whitney *U* test), and AuV (**** *p* < 0.0001, Mann–Whitney *U* test; Figure 4C, Table 2). Such salicylate-induced increases in the number of c-Fos-expressing cells were observed in both superficial and deep layers (Figure 4D).

When GJG was administered to salicylate-treated rats, the number of c-Fos-expressing cells was significantly reduced in the A1 (**** *p* < 0.0001, Mann–Whitney *U* test) and AuD (** *p* = 0.004, Mann–Whitney *U* test) but not in the AuV (*p* = 0.059 in SS + DW vs. SS + GJG, Mann–Whitney *U* test; Figure 4C). GJG-induced decreases in the numbers of c-Fos-expressing cells were observed in superficial and deep layers of the A1 and AuD (**** *p* < 0.0001 in A1 superficial, ** *p* = 0.0002 in A1 deep, * *p* = 0.021 in AuD superficial, *** *p* = 0.0008 in AuD deep, Mann–Whitney *U* test; Figure 4D).

In the auditory thalamus and brainstem, changes in the numbers of c-Fos-expressing cells were observed following salicylate and GJG administration (Figure 5, Table 3). In the MGB, salicylate injection significantly increased the number of c-Fos-expressing cells (** *p* = 0.008, Mann–Whitney *U* test), but GJG administration did not lead to a significant decrease in their number (*p* = 0.097, Mann–Whitney *U* test). In the DC and IC, GJG treatment caused a significant decrease in the number of c-Fos-expressing cells (* *p* = 0.016 in the DC, ** *p* = 0.010 in the ICc, Mann–Whitney *U* test), although salicylate injection did not significantly increase their number (*p* = 0.406 in the DC, *p* = 0.126 in the IC, Mann–Whitney *U* test).

## 4. Discussion

In the present study, we investigated the effects of GJG, which is clinically used for the palliative treatment of patients with tinnitus, and on behavioral and immunohistochemical parameters in salicylate-treated rats. GJG administration reduced the FP score and number of c-Fos-expressing neurons in rats. These results indicate that GJG is also effective for tinnitus treatment in animal models, and that this animal model is helpful in the study of its mechanisms and development of preventive/therapeutic interventions.

For the assessment of tinnitus induction, rats underwent the AA task using a protocol similar to that described in a previous study [16]. Comparing FP scores obtained from conditioning with different sound intensities and frequencies, Kizawa et al. [16] proposed that the conditioned stimulus of 60 dB and 16 kHz was the most appropriate for detecting tinnitus induction in rats that received 400 mg/kg salicylate for 3 days. Our present study showed no significant difference in FP scores at 40 dB and 60 dB although they were significantly higher than the scores at 80 dB. In addition, no significant differences in FP scores were found among 12-kHz, 16-kHz, and 20-kHz stimuli. These results suggest that the appropriate intensities and frequencies for the conditioned stimulus are somewhat broadly distributed. This finding supports those of previous reports, in which sound pressures of around 60 dB [27,28,29,30] and sound frequencies in the range of 10–16 kHz [9,11,30,31,32,33] were used for detection of tinnitus behaviors in salicylate-treated animals. An injection of salicylate significantly increased the FP score. However, the fact that high AA scores (≥ 80%) in salicylate-treated rats were comparable with those in control rats suggests that salicylate did not induce hearing loss in rats. Furthermore, a significant decrease in FP scores on the 4th day of the behavioral test session when salicylate was not applied to rats was demonstrated. This reversible effect of salicylate suggests that salicylate did not induce obvious functional damage to endocochlear organs. This reversible effect could be caused by a decreasing effect of cochlear blood flow [34].

This animal study was performed to objectively evaluate the effects of GJG on tinnitus. GJG was administered to rats at two different doses, 0.3 g/kg and 1.0 g/kg, which were higher than the daily dose clinically prescribed for patients (approximately 0.1 g/kg). However, these doses were also used in previous studies that evaluated the effects of GJG in a mouse model of streptozotocin-induced diabetes mellitus [23] and a rat model of oxaliplatin-induced acute peripheral neuropathy [22,24]. GJG administration at these doses caused a potent antinociceptive effect in the diabetic mice and prevented oxaliplatin-induced acute neuropathy, such as cold hyperalgesia and mechanical allodynia in the neuropathic rats. Therefore, we considered these doses appropriate for evaluating GJG effects in animal experiments. The salicylate-induced increase in FP score was significantly suppressed by administration of 1.0 g/kg GJG. This result suggests that GJG is effective for the suppression of tinnitus behavior in salicylate-treated rats.

Our present immunohistochemical study showed that the application of salicylate significantly increased the number of c-Fos-expressing cells in the primary and secondary auditory cortices and the MGB. These results correspond to findings of a previous study in gerbils [18]. In addition, our finding that the number of c-Fos-expressing cells was higher in superficial layers than in deep layers is in agreement with a previous result where c-Fos labeling peaks in layers 3 and 4 [20]. An enhancement of neural activity after salicylate application was also electrophysiologically demonstrated in the auditory cortex of cats and rats [30,35,36,37]. In auditory brainstem regions such as the DC and ICc, a significant increase in the number of c-Fos-expressing cells was not observed in salicylate-treated rats. This agrees with the salicylate-induced tinnitus model reported by Lanaia et al. [38]. They adopted the behavioral assessment of tinnitus using somatic resonance, where the DC played an important role. Their results indicate that the DC is not involved in salicylate-induced tinnitus but is involved in trauma-induced tinnitus. Our results are also consistent with those of Wu et al. [19], who found no increase in c-Fos in the DC, but not with those of Santos et al. [21], who found increases in the DC and ICc. These inconsistent findings may be attributed to different experimental conditions, such as different doses and periods of salicylate treatment, and time from SS administration to fixation (250 mg/kg salicylate for 5 days fixation after 8 h [19] and 300 mg/kg salicylate for 3 days fixation after 3 h [21], respectively).

When GJG was administered to salicylate-treated rats, significant decreases in the numbers of c-Fos-expressing cells were observed in the A1, AuD, DC, and ICc. A previous study has reported that salicylate application enhances functional connectivity of the auditory networks composed of the IC, MGB, and auditory cortices, in addition to nonauditory structures, such as the amygdala and cerebellum [39,40]. Our immunohistochemical results strongly suggest that brain regions such as the A1, AuD, DC, and ICc participate in salicylate-induced tinnitus.

Salicylate-induced tinnitus can be caused by peripheral and central effects. Because salicylate affects outer hair cells and auditory nerves [11,41,42,43], salicylate may have affected the peripheral auditory system, resulting in interactions with the central auditory system. Because salicylate can penetrate the blood–brain barrier [44,45,46], it has been reported that salicylate directly affects the central auditory system [39,47,48]. In vitro studies using brain slices have revealed that the application of salicylate modulates inhibitory synaptic responses and electrophysiological properties of central neurons [49,50,51,52,53,54]. Salicylate-induced tinnitus can be caused by peripheral and central effects.

A possible explanation for the suppressive effects of GJG on salicylate-induced tinnitus includes mechanisms via the transient receptor potential vanilloid 1 (TRPV1) channel [55]. TRPV1 channels are expressed in cochlear spiral ganglion cells, and the expression levels are upregulated by salicylate [16]. When TRPV1 channels are antagonized by a specific TRPV1 antagonist (capsazepine), salicylate-induced tinnitus behavior is suppressed [16]. A suppressive effect of GJG through TRP channels in oxaliplatin-induced acute neuropathy has also been reported [22]. Therefore, GJG might suppress FP scores and reduce the number of c-Fos-expressing cells through the modulation of TRPV1 channels. However, suppressive effects of GJG on TRPV1 in the cochlea have not been confirmed yet. Another possible explanation for the suppressive effects of GJG on salicylate-induced tinnitus includes the effects of increased peripheral blood flow. Dider et al. [34] reported that salicylate reduced cochlear blood flow. Kono et al. [56] reported that GJG ameliorated the oxaliplatin-induced decrease in peripheral blood flow. Therefore, GJG might suppress FP scores and reduce the number of c-Fos-expressing cells through increasing cochlear blood flow. However, increasing effects of GJG on blood flow in the cochlea have not been confirmed yet. On the other hand, GJG may directly act on central neurons. Several neuroprotective compounds of GJG (loganin, morroniside, catalpol, and paeoniflorin) were detected at high levels in the plasma of jugular vein blood shortly after oral GJG administration to rats [56]. This raises the possibility of a direct action of GJG on central neurons. In support of this notion, evidence suggesting direct GJG actions on central neurons is accumulating [56,57].

The limitations of this study are as follows. The rats used in our study were not old, because older rats have reduced memory and physical strength. It should be noted that this is not the same age at which tinnitus commonly occurs in humans. Although we examined only males in this study, the sex difference may affect behavioral tests or the auditory system [58]. However, there are reports that it can be used regardless of gender, even if the estrous cycle is present [59]. In any case, the effect of animal sex on the tinnitus behavioral model has not been examined. We performed immunohistochemical experiments and confirmed the suppression of c-Fos expression, but we did not evaluate the tinnitus behavior of these individuals as in the previous literature [16,21].

The question is whether our defined tinnitus behavior in rats really represent the presence of tinnitus. There is an experimental system that evaluated our FP (moved in the absence of stimuli) as a stress response. Depner et al. [60] reported an amplitude modulation study in which conditioning was performed by applying electrical stimuli in a shuttle box. Although not exactly the same conditioning, the behavior between sound stimuli was defined as inter-trial crossings (ITC), which are a measure of activity mainly describing the stress in animals. It is important to note that Depner’s study was not an experiment that examined tinnitus. Kizawa’s method [16] had already been reported as an established method for evaluating tinnitus behavior; thus, we believed that it would be possible to evaluate tinnitus by using this method. However, we cannot rule out the possibility that the FP scores we evaluated are affected by other factors such as stress in addition to active avoidance due to tinnitus.

In tinnitus evaluation, the gap detection test [6] has recently been adopted in a number of tinnitus behavioral experiments [38,61]. Since no conditioning was involved, unnecessary stress caused by conditioning may be less than that in our experiment. We evaluated tinnitus behavior using a method that can be validated at our facility, and we believe that Kizawa et al. ‘s method is sufficient to evaluate tinnitus.

## 5. Conclusions

We modeled tinnitus and demonstrated that GJG suppressed tinnitus behavior. We also demonstrated that GJG suppressed the increase in c-Fos expression by salicylate. Our study provided the first evidence that GJG suppressed tinnitus, which had never been presented in animal studies. The results provide evidence for future clinical trials of GJG. We hope that this study will help establish a clinical treatment for tinnitus.

## Figures and Tables

**Figure 1 brainsci-12-00587-f001:**
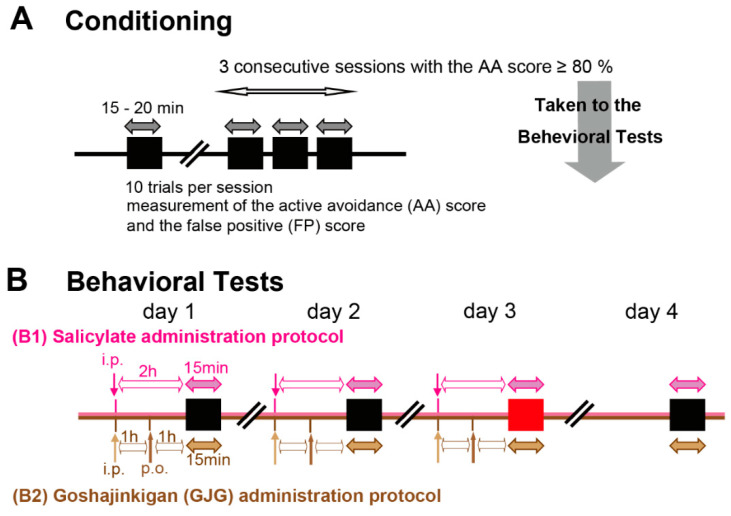
Schematic representation of the behavioral test protocol. (**A**) Conditioning protocol. Animals were conditioned to move to the opposite chamber in response to a sound presentation. Each session of 10 trials lasted 15–20 min. When conditioned (criterion: three consecutive sessions with an active avoidance (AA) score ≥ 80%), animals underwent the behavioral tests. (**B**) Behavioral tests. In the salicylate administration protocol (B1), salicylate was intraperitoneally injected 2 h before the behavioral measurement (i.p., pink arrow). In this behavioral test, the correct movements to sound (AA score) and the movements during intertrial periods without sound (false positive (FP) score) were measured in a 15 min session. The drug was intraperitoneally injected daily for three days, and the behavioral test was performed daily for four days. In the goshajinkigan (GJG) administration protocol (B2), salicylate was intraperitoneally injected 2 h before the behavioral measurement (i.p., light brown arrow). Distilled water or GJG was orally administered (p.o., dark brown arrow) 1 h before the behavioral measurement. For the control group, animals received daily i.p. saline injections and following oral administration of DW. The i.p. and p.o. administrations were performed daily for three days, and the behavioral test was performed daily for four days.

**Figure 2 brainsci-12-00587-f002:**
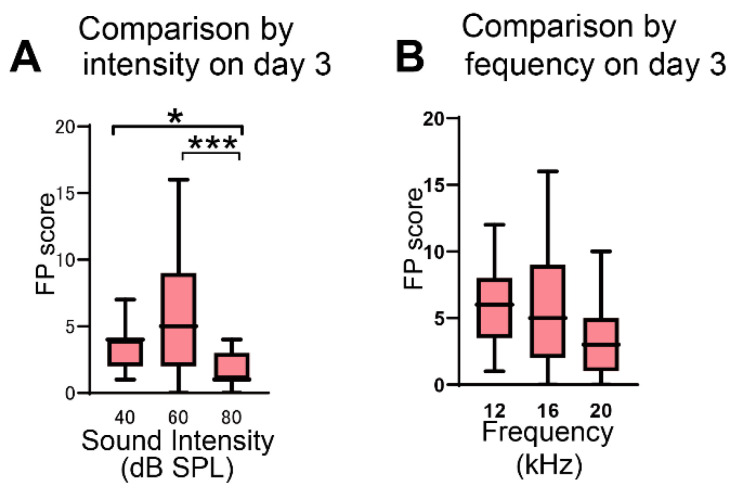
Verification of the sound intensity and frequency in salicylate-treated rats. (**A**) Comparison of FP scores at 16 kHz in three groups conditioned with sound pressures of 40 dB (*n* = 19), 60 dB (*n* = 31), and 80 dB (*n* = 12). The FP scores at 40 dB and 60 dB were significantly higher than those at 80 dB (* *p* = 0.049 in 40 dB vs. 80 dB, *** *p* = 0.0003 in 60 dB vs. 80 dB, Kruskal–Wallis test and Dunn’s test). (**B**) Comparison of FP scores at 60 dB in three groups conditioned with sound frequencies of 12 kHz (*n* = 17), 16 kHz (*n* = 31), and 20 kHz (*n* = 17). No significant difference was observed among these three groups. Data are presented as median (interquartile range).

**Figure 3 brainsci-12-00587-f003:**
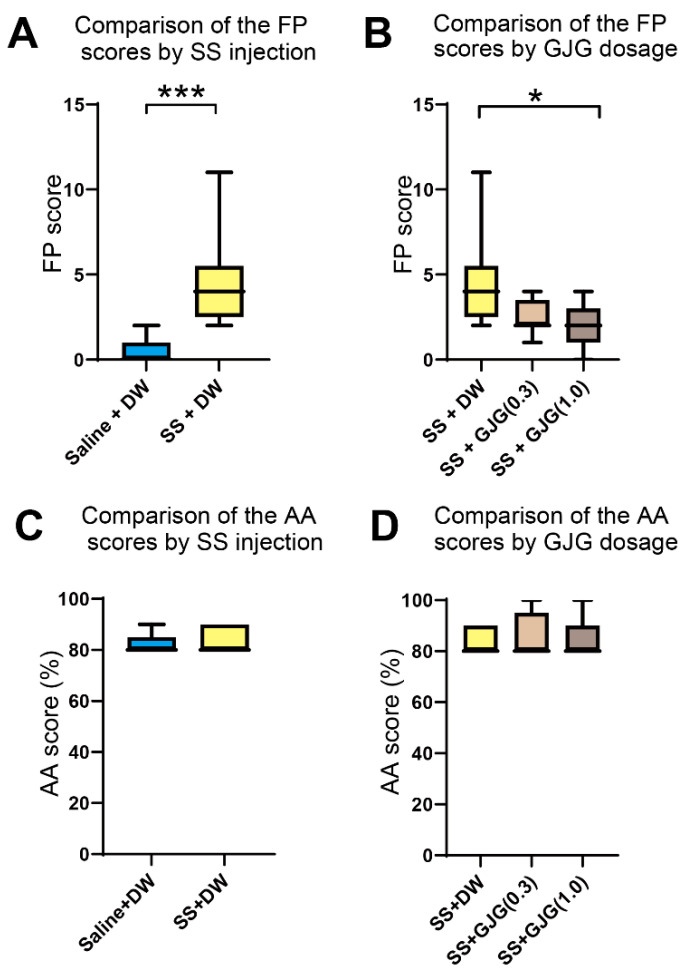
Effect of goshajinkigan (GJG) administration on false positive (FP) and active avoidance (AA) scores in salicylate-treated rats. (**A**) Comparison of FP scores between Saline + distilled water (DW) group and sodium salicylate (SS) + DW group (*n* = 9 in each group). The FP scores in the SS + DW group were significantly higher than those in the Saline + DW group (*** *p* = 0.0002, Mann–Whitney *U* test). (**B**) Comparison of FP scores among three groups: SS + DW group, SS + GJG (0.3) group, and SS + GJG (1.0) group (*n* = 9 in each group). The FP scores in the SS + GJG (1.0) group were significantly lower than those in the SS + DW group (* *p* = 0.021, Kruskal–Wallis test and Dunn’s test). No significant difference was observed between the SS + DW and SS + GJG (0.3) (*p* = 0.275, Kruskal–Wallis test and Dunn’s test) or SS + GJG (0.3) and SS + GJG (1.0) groups (*p* = 0.948, Kruskal–Wallis test and Dunn’s test). (**C**) Comparison of AA scores between the Saline + DW group and SS + DW group. Both groups showed AA scores ≥ 80% without significant differences. (**D**) Comparison of AA scores in three groups: SS + DW group, SS + GJG (0.3) group, and SS + GJG (1.0) group. All groups showed AA scores ≥ 80% with no significant group differences. Data are presented as median (interquartile range). A learning curve was created to show the progression of AA scores until these groups completed conditioning (see Appendix A).

**Figure 4 brainsci-12-00587-f004:**
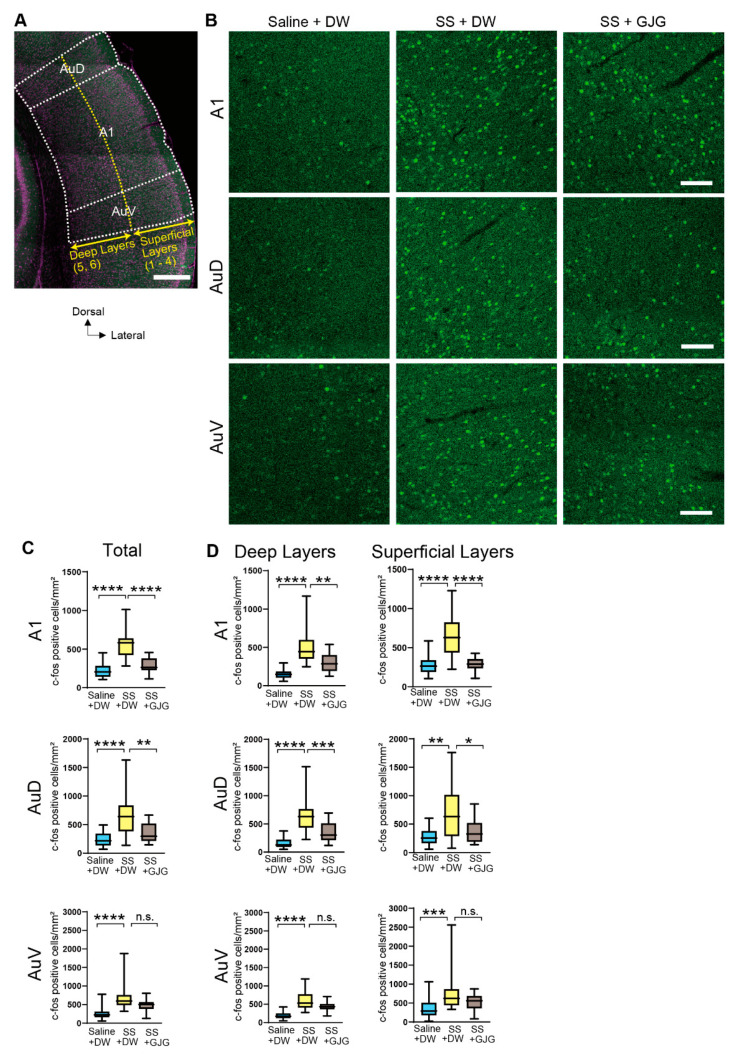
Effect of goshajinkigan (GJG) administration on c-Fos expression in the auditory cortex of salicylate-treated rats. (**A**) A low magnification fluorescent photomicrograph of the auditory cortex. The primary (A1) and dorsal (AuD) and ventral (AuV) secondary auditory cortexes were separated based on Nissl staining (shown in magenta). Layers 1–4 and 5–6 were defined as superficial and deep layers, respectively (yellow dotted lines). Scale bar = 500 μm. (**B**) Fluorescent photomicrographs in the areas of the A1, AuD, and AuV. Green dots show c-Fos-expressing cells in the Saline + distilled water (DW) group (left), sodium salicylate (SS) + DW group (center), and SS + GJG group (right). Scale bars = 100 μm. (**C**) Comparison of the numbers of c-Fos-expressing cells in the Saline + DW, SS + DW, and SS + GJG groups. In the A1 and AuD, cell numbers in the SS + DW group were significantly higher than those in the Saline + DW and SS + GJG groups. In the AuV, the number in the SS + DW group was significantly higher than the number in the Saline + DW group. (**D**) Comparison of the numbers of c-Fos-expressing cells in the superficial and deep layers. In both superficial and deep layers of the A1 and AuD, the numbers in the SS + DW group were significantly higher than those in the Saline + DW and SS + GJG groups. In the superficial and deep layers of the AuV, the numbers in the SS + DW group were significantly higher than those in the Saline + DW group. Data are presented as median (interquartile range) and are analyzed using the Mann–Whitney *U* test, *n* = 6 in each group. Asterisks indicate a significant difference between groups (* *p* < 0.05; ** *p* < 0.01; *** *p* <0.001; **** *p* <0.0001).

**Figure 5 brainsci-12-00587-f005:**
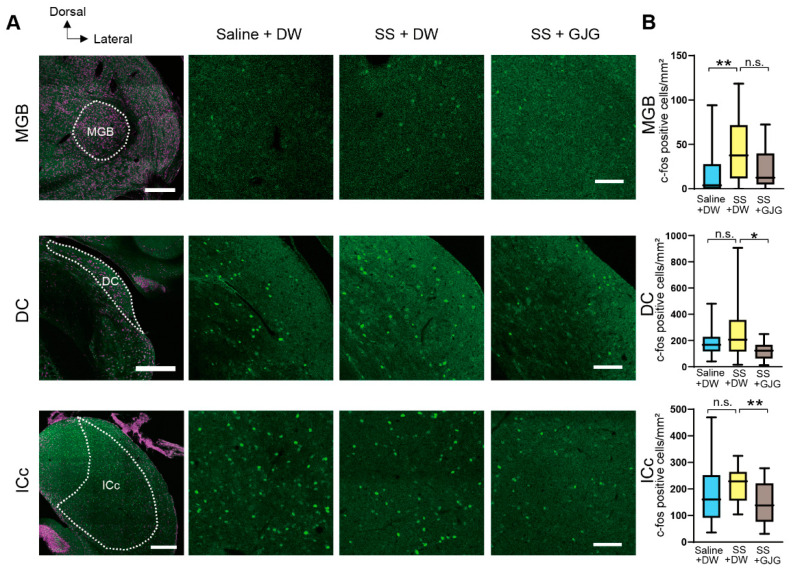
Effect of goshajinkigan (GJG) administration on c-Fos expression in the auditory thalamus and brainstem nuclei of salicylate-treated rats. (**A**) Fluorescent photomicrographs of the medial geniculate body (MGB), dorsal cochlear nucleus (DC), and central nucleus of the inferior colliculus (ICc). These areas were separated based on Nissl stainings (shown in magenta) in low magnification fluorescent photomicrographs (leftmost column). Green dots show c-Fos-expressing cells in the Saline + distilled water (DW) group (second column), sodium salicylate (SS) + DW group (third column), and SS + GJG group (rightmost column). Scale bars = 500 μm (leftmost column) and 100 μm (rightmost column). (**B**) Comparison of the numbers of c-Fos-expressing cells in Saline + DW, SS + DW, and SS + GJG groups. In the MGB, the number of c-Fos-expressing cells was significantly larger in the SS + DW group than in the Saline + DW group. In the DC and ICc, cell numbers in the SS + DW group were significantly higher than those in the SS + GJG group. Data are presented as median (interquartile range) and are analyzed using the Mann–Whitney *U* test, *n* = 6 in each group. Asterisks indicate a significant difference between groups (* *p* < 0.05; ** *p* < 0.01).

**Table 1 brainsci-12-00587-t001:** Comparison of FP scores on days 3 and 4.

		With SS	Without SS	
Intensity (SPL)	Frequency (Hz)	FP on Day 3 Median (IQR)	FP on Day 4 Median (IQR)	*p* Value
40 dB	16 k	4.0 (2.0–4.0)	1.0 (0.0–3.0)	** *p* = 0.0013
60 dB	16 k	5.0 (2.0–9.0)	2.0 (1.0–4.0)	**** *p* < 0.0001
80 dB	16 k	1.0 (1.0–3.0)	0.0 (0.0–1.0)	* *p* = 0.0313
60 dB	12 k	6.0 (3.5–8.0)	1.0 (0.5–5.0)	** *p* = 0.0065
60 dB	20 k	3.0 (1.0–5.0)	1.0 (0.0–2.0)	** *p* = 0.0076

Scores on days 3 and 4 are compared, and all show a significant decrease in the Wilcoxon test. Data are presented as median [interquartile range (IQR) 25–75]. FP: false positive, SPL: sound pressure level, SS: sodium salicylate. Asterisks indicate a significant difference between groups (* *p* < 0.05; ** *p* < 0.01; **** *p* <0.0001).

**Table 2 brainsci-12-00587-t002:** Effect of goshajinkigan (GJG) administration on c-Fos expression in the auditory cortex of salicylate-treated rats.

Area	Group	c-Fos Expressing Cells Median (IQR)	*p* Value
	Saline + DW	205 (143–286)	**** *p* < 0.0001 **** *p* < 0.0001
A1 total	SS + DW	583 (442–641)	
	SS + GJG	264 (227–381)	
	Saline + DW	218 (134–343)	**** *p* < 0.0001 ** *p* = 0.004
AuD total	SS + DW	641 (382–837)	
	SS + GJG	297 (214–518)	
	Saline + DW	221 (157–308)	**** *p* < 0.0001 *p* = 0.059
AuV total	SS + DW	595 (484–760)	
	SS + GJG	499 (382–567)	
	Saline + DW	264 (187–339)	**** *p* < 0.0001 **** *p* < 0.0001
A1 superficial	SS + DW	630 (436–825)	
	SS + GJG	290 (235–355)	
	Saline + DW	255 (164–378)	** *p* = 0.002 * *p* = 0.021
AuD superficial	SS + DW	632 (288–1013)	
	SS + GJG	328 (191–521)	
	Saline + DW	290 (176–509)	*** *p* = 0.0007 *p* = 0.214
AuV superficial	SS + DW	624 (439–869)	
	SS + GJG	561 (362–686)	
	Saline + DW	149 (104–186)	**** *p* < 0.0001 ** *p* = 0.002
A1 deep	SS + DW	445 (352–599)	
	SS + GJG	286 (192–401)	
	Saline + DW	127 (92–221)	**** *p* < 0.0001 *** *p* = 0.0008
AuD deep	SS + DW	630 (431–764)	
	SS + GJG	301 (212–511)	
	Saline + DW	171 (122–253)	**** *p* < 0.0001 *p* = 0.069
AuV deep	SS + DW	531 (408–777)	
	SS + GJG	434 (364–509)	

The number of c-Fos-expressing cells are compared. Mann–Whitney *U* test. A1: primary auditory cortex, AuD: secondary auditory cortex dorsal zone, AuV: secondary auditory cortex ventral zone, SS: sodium salicylate, DW: distilled water, IQR: interquartile range. Asterisks indicate a significant difference between groups (* *p* < 0.05; ** *p* < 0.01; *** *p* <0.001; **** *p* <0.0001).

**Table 3 brainsci-12-00587-t003:** Effect of goshajinkigan (GJG) administration on c-Fos expression in the auditory thalamus and brainstem nuclei of salicylate-treated rats.

Area	Group	c-Fos Expressing Cells Median (IQR)	*p* Value
	Saline + DW	4 (0–28)	** *p* = 0.008 *p* = 0.097
MGB	SS + DW	38 (12–72)	
	SS + GJG	13 (5–40)	
	Saline + DW	167 (117–229)	*p* = 0.406 * *p* = 0.016
DC	SS + DW	206 (116–357)	
	SS + GJG	123 (62–168)	
	Saline + DW	161 (91–253)	*p* = 0.126 ** *p* = 0.010
ICc	SS + DW	229 (156–264)	
	SS + GJG	138 (76–221)	

The number of c-Fos-expressing cells are compared. Mann–Whitney *U* test. MGB: medial geniculate body, DC: dorsal cochlear nucleus, ICc: central nucleus of inferior colliculus, SS: sodium salicylate, DW: distilled water, IQR: interquartile range. Asterisks indicate a significant difference between groups (* *p* < 0.05; ** *p* < 0.01).

## Data Availability

The data that support the findings of this study are available from the first author, K.K., upon reasonable request.

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
