# Peer review of "Behavioral and Immunohistochemical Evidence for Suppressive Effects of Goshajinkigan on Salicylate-Induced Tinnitus in Rats"

_brainsci, 2022, doi:10.3390/brainsci12050587_

Round 1
Reviewer 1 Report
The present study models tinnitus in rats using salicylate and demonstrates the benefits of GJG a popular traditional Japanese medicine prescribed for people with this disorder. Probably, the most relevant part of the study is the use of the model to provide an immunohistochemical analysis that points at specific brain regions that participate in salicylate-induced tinnitus and can be established as a starting point for future research to understand the mecanisms. This part is not, but should be reflected in the title since it can be considered a major contribution that is hidden. In the next paragraphs, some other aspects are indicated and discussed in a constructive manner.
Title – Please, include the findings of immunohistochemistry, as this is a contribution of the model.
Abstract- The abstract should be densified in the contents related to the relevance / need to have an animal model, the possibility to find neuroanatomical substrates and be used in future experiments. For instance… “ no evidence is provided by animal studies” is a flattened argument. Please, reorganize the sentence / so it is clear that ‘its behavioral effects and the underlying neuroanatomical substrate haven’t been modeled in animals’ or something similar that makes clear the need of use of animals when the drug is already known as effective. Since the behavioral paradigm uses avoidance, this translational approach needs to be emphasized. The abstract and title, should clearly indicate the three (in my opinion) findings: First, you are able to model the tinnitus, second, you demonstrate the effectiveness of GJG reversing the effects of salicylate, third, your c-fos experiments show the underlying neuronal substrate of the most effective dose.
The authors refer to prevalence in the general population, and also provide that in adult japanese people. Since the purpose is to provide a modeling in rats, please, indicate in the discussion the translation of the age of animals into humans. The age of animals (8-23 PNW) chosen was suitable to mimic which part of the population?.
For objective evaluation of tinnitus, animal models 34 of tinnitus can be useful.
please, also add “ For objective evaluation of tinnitus, “”””” the study of its mechanisms and development of preventive/therapeutical interventions,”””” animal models 34 of tinnitus can be useful.” As it is also referred somewhere else in the text but it is pertinent here, at the beginning, mostly since some reports show that GJG has been effective in humans.
In a similar manner, please do so in lines 309-310 when the authors refer to “These results indicate that GJG is also effective for 309 tinnitus treatment in animal models.” and…
Methods
Since the behavioral procedure implies electric shocks, and this is an extremely sensitive issue for animal welfare, please, clarify (i.e. lines 77-78) that the paradigm is an escapable paradigm or avoidance task and the mA are not harmful since a maximum of 30 seconds is not an instant, and it is expected that the model will imply hearing loss, so more exposure times for the ‘rats’ modeling tinnitus.
Why did the authors not use the animals behaviourally assessed to run the immunos? That would allow to do functional correlation analysis.
The rationale for using the highest dose (1.0 g/kg) should be given in section 2.3.
The reversible salicylate effect in this behavioral test seems an important issue and, as the authors indicate with supporting references 9 and 16, has been previously reported. Also, as discussed in lines 323 325 that, AA scores in treated rats were similar to controls. To which extent this model the effects of salicylate in humans? Can it be relevant here? Is it a limitation? Can a final sentence on line 328 make a conclusion on what scenario is modeled?
The sentence about both central and peripheral effects would be better located at the beginning of the paragraph (in line 361), better than at the end (367).
Limitations of the study with respect to only using males and not both sexes should be included also adding a comment on the expectations in the female sex.
Instead of sentence 382, a short paragraph of all the results as a conclusion, also including further directions, is necessary.
Author Response
Response to reviewer 1 Comments
Point 1: Title – Please, include the findings of immunohistochemistry, as this is a contribution of the model.
Response 1: Thank you for your suggestion. We added “Behavioral and Immunohistochemical Evidence for” in the title. I hope the title becomes easier for understanding.
Point 2: Abstract- The abstract should be densified in the contents related to the relevance / need to have an animal model, the possibility to find neuroanatomical substrates and be used in future experiments. For instance… “ no evidence is provided by animal studies” is a flattened argument. Please, reorganize the sentence / so it is clear that ‘its behavioral effects and the underlying neuroanatomical substrate haven’t been modeled in animals’ or something similar that makes clear the need of use of animals when the drug is already known as effective. Since the behavioral paradigm uses avoidance, this translational approach needs to be emphasized. The abstract and title, should clearly indicate the three (in my opinion) findings: First, you are able to model the tinnitus, second, you demonstrate the effectiveness of GJG reversing the effects of salicylate, third, your c-fos experiments show the underlying neuronal substrate of the most effective dose.
Response 2: Thank you for your suggestion. We changed “no evidence is provided by animal studies” to “its behavioral effects and the underlying neuroanatomical substrate have not been modeled in animals”. We changed “Using behavioral and histological analyses in this study, we investigated the effects of GJG on salicylate-treated rats, which showed behavior indicative of tinnitus” to “We modeled tinnitus using salicylate-treated rats, demonstrated the effectiveness of GJG on tinnitus, and examined the underlying neuronal substrate with c-Fos expression”. (lines in 18-21)
Point 3: The authors refer to prevalence in the general population, and also provide that in adult japanese people. Since the purpose is to provide a modeling in rats, please, indicate in the discussion the translation of the age of animals into humans. The age of animals (8-23 PNW) chosen was suitable to mimic which part of the population?.
Response 3: Thank you for your question. The age of animals examined in this study corresponded to the age of adolescence or young adulthood in humans. Because of the difficulty of conditioning in older animals due to their reduced memory and physical strength, we adapted animals of this age were used. The age of animals we used was not suitable to mimic to the human population. This was additionally noted in the limitation (lines 415-417). “The rats used in our study were not old, because older rats have reduced memory and physical strength. It should be noted that this is not the same age at which tinnitus commonly occurs in humans..”
Point 4: For objective evaluation of tinnitus, animal models 34 of tinnitus can be useful.
please, also add “ For objective evaluation of tinnitus, “”””” the study of its mechanisms and development of preventive/therapeutical interventions,”””” animal models 34 of tinnitus can be useful.” As it is also referred somewhere else in the text but it is pertinent here, at the beginning, mostly since some reports show that GJG has been effective in humans.
Response 4: We added your suggestion (in lines 36-37). Thank you very much.
Point 5: In a similar manner, please do so in lines 309-310 when the authors refer to “These results indicate that GJG is also effective for 309 tinnitus treatment in animal models.” and…
Response 5: Thank you for your suggestion. We changed “These results indicate that GJG is also effective for tinnitus treatment in animal models.” to “These results indicate that GJG is also effective for tinnitus treatment in animal models, and this animal model is helpful for the study of its mechanisms and development of preventive/therapeutic interventions.”
Point 6: Methods- Since the behavioral procedure implies electric shocks, and this is an extremely sensitive issue for animal welfare, please, clarify (i.e. lines 77-78) that the paradigm is an escapable paradigm or avoidance task and the mA are not harmful since a maximum of 30 seconds is not an instant, and it is expected that the model will imply hearing loss, so more exposure times for the ‘rats’ modeling tinnitus.
Response 6: Thank you for your suggestion. Experimental methods were based on the well-established method of Kizawa et al. The same 3.7 mA electrical stimulation was used in other experimental systems (Guitton, et al. 2003), and most rats were moved to an adjacent chamber before the 30 s electrical stimulation. Electrical stimulation at 3.7 mA for 30 seconds did not cause the rats to shiver and become immobile.
We added “The paradigm was so escapable that rats easily moved to the next chamber.” (in line 81).
Point 7: Why did the authors not use the animals behaviourally assessed to run the immunos? That would allow to do functional correlation analysis.
Response 7: Thank you for your suggestion.
First, the behavioral experiment of the GJG administration was originally planned to include the fourth day of the behavioral experiment by Kizawa et al. to see reversible effects. After the behavioral experiments were closed, immunohistochemical experiments were planned and conducted.
Second, other tinnitus experiments do not always include histological evaluation of animals that exhibit tinnitus behavior (Kizawa et al. 2010, Santos et al. 2017).
Therefore, we did not perform behavioral experiments for the immunohistochemistry experiments, and we describe this as a limitation. (in lines 422-423)
Point 8: The rationale for using the highest dose (1.0 g/kg) should be given in section 2.3.
Response 8: Thank you for your suggestion.
There are two reasons for using the highest dose (1.0 g/kg).
First, animal studies using GJG often use a 1.0 g/kg dose. (Mizuno, K. et al. 2014; Kono, T. et al. 2015; Hagihara, K. et al. 2022)
Second, because the results of the behavioral experiment (3.2) showed a line significant effect, we adapted 1.0g/kg for the immunohistochemical study.
It seems inappropriate to list this amount even though the results are not yet available, so it is not listed.
Point 9: The reversible salicylate effect in this behavioral test seems an important issue and, as the authors indicate with supporting references 9 and 16, has been previously reported. Also, as discussed in lines 323 325, AA scores in treated rats were similar to controls. To which extent this model the effects of salicylate in humans? Can it be relevant here? Is it a limitation? Can a final sentence on line 328 make a conclusion on what scenario is modeled?
Response 9: Thank you for your question. For the Reversible effect, Dider, A., Miller, J.M., and Nuttall, A.L. (1993) reported that salicylate administration decreased cochlear blood flow in guinea pigs. We consider the possibility that a temporary increase and decrease in cochlear blood flow caused by salicylate may contribute to the Reversible effect. Since the Reversible effect was not mentioned in the Discussion section, we added this point to the Discussion.
From this perspective, the peripheral blood flow increasing effects of GJG (Kono, T. et al. 2015)[54] may have reduced tinnitus behavior and c-fos expression. We added it to the Discussion (in lines 403-409).
In our study, salicylate was administered for 3 days, which can be taken as a model of acute tinnitus. Experiments on chronic tinnitus with salicylate administered for 14 days have already been conducted, which is a future problem for us. (Wu, C., Wu, X., Yi, B., et al. Am J Transl Res. 2018)
Point 10: The sentence about both central and peripheral effects would be better located at the beginning of the paragraph (in line 361), better than at the end (367).
Response 10: Thank you for your suggestion. The sentence you pointed out has been moved to the beginning of the paragraph. It is now easier to understand.
Point 11: Limitations of the study with respect to only using males and not both sexes should be included also adding a comment on the expectations in the female sex.
Response 11: Thank you for your suggestion. We added a limitation at the end of our discussion, adding, " Although we examined only males in this study, the sex difference may affect behavioral tests or the auditory system [58]. However, there are reports that it can be used regardless of gender, even if the estrous cycle is present [59]. In any case, the effect of animal sex on the tinnitus behavioral model has not been examined..”(in lines 417-421)
Point 12: Instead of sentence 382, a short paragraph of all the results as a conclusion, also including further directions, is necessary.
Response 12: Thank you for your suggestion. We added the “5. Conclusion” paragraph and added “We modeled tinnitus and demonstrated that GJG suppressed tinnitus behavior. We also demonstrated that GJG suppressed the increase in c-Fos expression by salicylate. Our study provided the first evidence that GJG suppressed tinnitus, which had never been presented in animal studies. The results provide evidence for future clinical trials of GJG. We hope that this study will help establish a clinical treatment for tinnitus.” (in lines 442-446)

Reviewer 2 Report
Suppressive Effects of Goshajinkigan on Salicylate-Induced Tinnitus in Rats
The authors aim to establish a new model for investigating the effects of the traditional Japanese medicine Goshajinkigan (GJG) on tinnitus. It is known that GJG has positive effects in tinnitus patients but for future research, an animal model is missing yet. They propose a shuttle box tone detection paradigm – even though, they are calling the setup a “conditioning box” – with counting the inter trial jumps as “false positive (FP)” indications of a tinnitus percept after 400 mg/kg body weight (BW) salicylate injections, while using the conditioned responses to pure tones (“active avoidance, AA”) as an indirect measurement for hearing loss. They find that high doses of orally applied GJG (1g/kg BW) significantly reduces the salicylate-induced increase of the FP rate. Additionally, the authors investigate the c-Fos expression in several auditory brain areas and find that high doses of GJG prevent the salicylate related c-Fos increase. They conclude that the proposed animal model can be used to investigate the effects of GJG in human tinnitus patients.
General comments:
I think that the proposed model might be useful for the investigation of salicylate-induced tinnitus or cortical effects of the tinnitus percept in general and the treatment with GJG in patients, but not so much for the understanding of the development of tinnitus – which is not in the focus of the authors. Nevertheless, the manuscript has several flaws that need to be addressed before it can be endorsed for publication.
First of all the Methods section has to be rewritten in large parts. Especially the sections regarding the groups and the shuttle box paradigm. I had huge problems to figure out how many animals were investigated with which paradigm and how the treatment was done exactly. For example, the authors state that they investigated 141 animals, but explain only 54 of them in the Methods; the remaining 87 animals suddenly appear in the Results section. The paradigms have to be explained exactly and not be distributed over several subsections with adding important information bit by bit. The information in Figure 1 is not in line with the description in the text. E.g., the control animals are missing completely and the salicylate treated animals received water orally at the same times the GJG animals received their treatment. I suggest to fuse the subsections 2.2.1 to 2.2.5 into one chapter and explain also all test groups in detail. Additionally, I would like to see the learning curves for the different groups to get a feeling for the difficulty of the trained task. This can be added as a supplement.
Second, in the statistics used, I have the impression of confusion. Sometimes the authors use non-parametric and sometimes parametric statistics. They claim to use parametric statistics only when the distribution of the data allows it, but this seems to be only the case in the AA data and there, no significant differences could be found.
The authors should only use non-parametric statistics to reduce confusion. In this context, the authors have to redo most of their figures. As mentioned, most of the time they use non-parametric statistics but always shoe the mean and SEM in the figures. Please use the median and the interquartile interval (25-75% interqantile interval) for all figures. Then also the statistics matches the data shown.
Third, in the Discussion section, I miss a critical discussion of the behavioral approach used. The authors explain, that in the test animal groups the FP rate was higher in 40 and 60 dB conditions compared to the 80 dB condition (lines 315ff) while different frequencies around beast hearing did not differ. The first observation could indicate that the animals did not react (only) to tinnitus but were more stressed, when the task was more difficult. The FP the authors are counting are usually seen as inter trial jumps or inter trial counts (ITC) which indicate the stress level of the animals (e.g., Depner, M., Tziridis, K., Hess, A., and Schulze, H. (2014) or Schulze, H., and Tziridis, K. (2015)). The second observation in the context of stress (but not tinnitus) makes sense, as the frequencies are all well in the area of best hearing. Nevertheless, tinnitus and stress might be closely related and needs to be discussed. Additional points for the discussion are given in the specific comments below.
Specific comments:
Abstract
15 is “Kampo” needed here?
19 change “behavior indicative” to “”behavioral correlates”
20 for three consecutive days
21 after each salicylate injection
Introduction
37 “…various behavioral tests [6]” you can also add Turner et al (2006) here and mention that this is not a conditioned response test but a reflex based test.
48 For the patients it would be nice to know, how fast and how long the effect was, that has been observed. This is also important for the time you choose for your experiments.
Methods
69 For the time of day, can you provide information when this was relative to light off? Was this also the case during test conditions?
70ff As mentioned above, the whole section 2.2 should be rewritten and Fig. 1 should be also revised. Nevertheless, I’ll provide the comments to important information that is missing in the section.
75 the standard term for the “conditioning box” is shuttle box
80 define pure tone frequencies
81f the footshock started 1s after the end of the sound stimulus
85 change “section” to “session”
86 When was the response of the animal in the Shuttle box counted as correct? Please explain this here and not in the next paragraph. I thought that if you have a inter trial time (ITT) of 1 min and a response time of just 1 sec the animal has hardly any chance to avoid the shock completely. You see, that I completely misunderstood the paradigm initially!
89 FP: this is only counted in the 1 min ITTs. You write it in the figure caption but not here
91 80%: how long did the training take. Can you provide learning curves?
Figure 1: B and C can be united; the control group (saline) is missing completely and as I understood all animals received something (water/GJG) orally.
138 Explain, that these are different animals.
173ff As mentioned above, change the statistics accordingly.
Results
Table 1: also here use median and 25-75% values
191ff: As mentioned above, please change the statistics where necessary.
196 Why do you start with Fig. 2C? Is it possible to rearrange the order of the figure or the order in the text?
199f “The FP scores…” you just said that they were not different. Please clarify.
203 Which FP scores are investigated?
Figure 2: separate the Figure into two, as A-C investigate something completely different than D-F. In line 214 you show the remaining animal groups for the first time!
233 again, which FP scores are investigated?
238 No mentioning of Figure 2E in the results?
249 Why do you use these tests? Why not a Kruskal-Wallis ANOVA with post-hoc tests? If you use MW U tests, did you correct for multiple testing (e.g. Bonferroni)?
260 why a t test? Anyway, do not use parametric tests.
Figure 3 and 4: Remove the statistics from the figure captions. Put them in separate tables, use median and 25-75%.
Discussion
316f 40 vs 60 vs 80 dB: As mentioned above, this could indicate, that the animals did not react to tinnitus but were more stressed, when the task was more difficult. The FP you are counting are usually seen as ITCs which indicate the stress level of the animals. Please discuss this.
318 12-kHz, 16-kHz, and 20-kHz stimuli: in the context of stress (but not tinnitus) this makes sense as the frequencies all are well in the area of best hearing. Nevertheless, tinnitus and stress might be closely related. But is needs to be discussed.
325 hearing loss: at least not in the range that affects the detection. See Lanaia et al. (2021)
327 functional damage to endocochlear organs: did you look for bleedings? A high dose of salicylate might induce them. In gerbils, doses of "only" 300 mg/kg sometimes lead to bleedings in the cochlea.
331 patients: is it known, if GJG has stress/anxiety releasing effects in humans? If FP is - at least partially - a stress indicator, it could explain parts of the effect. Else, it would point to a more tinnitus related effect.
339 1 g/kg GJG: have these concentrations ever been tested in humans? They might have other/additional effects than the low doses (in animals factor 10 is quite high compared to humans, most of the time, factor 3 (your low dose) is more common)
353 maybe mention what the exact differences of the two mentioned studies compared to your study are.
362f “central auditory system”: here you could discuss the effects found in Lanaia et al. 2021
382 The statement in your concluding sentence was already known. You showed that your model might be capable of being used for the investigation of the described effect.
Author Response
General comments
Point 1: First of all the Methods section has to be rewritten in large parts. Especially the sections regarding the groups and the shuttle box paradigm. I had huge problems to figure out how many animals were investigated with which paradigm and how the treatment was done exactly. For example, the authors state that they investigated 141 animals, but explain only 54 of them in the Methods; the remaining 87 animals suddenly appear in the Results section. The paradigms have to be explained exactly and not be distributed over several subsections with adding important information bit by bit. The information in Figure 1 is not in line with the description in the text. E.g., the control animals are missing completely and the salicylate treated animals received water orally at the same times the GJG animals received their treatment. I suggest to fuse the subsections 2.2.1 to 2.2.5 into one chapter and explain also all test groups in detail. Additionally, I would like to see the learning curves for the different groups to get a feeling for the difficulty of the trained task. This can be added as a supplement.
Response 1: Thank you very much for your suggestion.
First, the number of animals used was noted at the beginning of each paragraph of the experiment. I apologize for the total number of animals. Perhaps I incorrectly counted 3 groups of 9 animals each in the behavioral tests with GJG for a total of 141 animals, but in fact, there were 4 groups of 9 animals each for a total of 150 animals. I apologize for the extra confusion.
Since the control group was not explained in the text, we added “For the control group (n = 9), rats received daily i.p. saline injections following oral administration of DW.” (in lines 128-129)
We fused the subsections 2.2.1 to 2.2.5 into one chapter and explain all test groups in detail.
We made learning curves for behavioral tests following GJG administration groups and added as supplementary Figure 1.
Point 2: Second, in the statistics used, I have the impression of confusion. Sometimes the authors use non-parametric and sometimes parametric statistics. They claim to use parametric statistics only when the distribution of the data allows it, but this seems to be only the case in the AA data and there, no significant differences could be found.
The authors should only use non-parametric statistics to reduce confusion. In this context, the authors have to redo most of their figures. As mentioned, most of the time they use non-parametric statistics but always shoe the mean and SEM in the figures. Please use the median and the interquartile interval (25-75% interqantile interval) for all figures. Then also the statistics matches the data shown.
Response 2: Thank you for your suggestion about the statistics. We used parametric tests for the AA data and the c-Fos expression data in MGB, but since nonparametric tests gave the same results, we decided to unify them with nonparametric tests. We rewrote the results of the "2.5. Statical analysis" and changed the test results to re-calculated results with nonparametric tests.
We changed the median and the interquartile interval (25-75% interquartile, interval) for all figures.
Point 3: Third, in the Discussion section, I miss a critical discussion of the behavioral approach used. The authors explain, that in the test animal groups the FP rate was higher in 40 and 60 dB conditions compared to the 80 dB condition (lines 315ff) while different frequencies around beast hearing did not differ. The first observation could indicate that the animals did not react (only) to tinnitus but were more stressed, when the task was more difficult. The FP the authors are counting are usually seen as inter trial jumps or inter trial counts (ITC) which indicate the stress level of the animals (e.g., Depner, M., Tziridis, K., Hess, A., and Schulze, H. (2014) or Schulze, H., and Tziridis, K. (2015)). The second observation in the context of stress (but not tinnitus) makes sense, as the frequencies are all well in the area of best hearing. Nevertheless, tinnitus and stress might be closely related and needs to be discussed. Additional points for the discussion are given in the specific comments below.
Response 3: Thank you for your suggestion. We were missing a discussion about the possibility of assessing stress rather than tinnitus in our behavioral approach.
As you pointed out, stress needs to be discussed. In behavioral experiments that do not require conditioning such as the interquartilegap detection test used by Lanaia, V. et al. (2021) and Schulze, H., and Tziridis, K. (2015), there may be less need to consider stress effects than in that require conditioning.
Unfortunately, our facility did not have the equipment for this gap detection test, nor did we have the experts to handle it. We decided to evaluate tinnitus using the behavioral experiment of Kizawa et al. (2010), which could be replicated at our facility.
This method has already been reported as a method for evaluating tinnitus behavior, and we thought it would be possible to evaluate tinnitus.
Although we cannot completely rule out the possibility of stress-induced chamber shifts, we believe that we assessed tinnitus behavior.
With the papers presented to us, we added the following sentences:
“The question is whether our defined tinnitus behavior in rats really represent the presence of tinnitus. There is an experimental system that evaluated our FP (moved in the absence of stimuli) as a stress response. Depner et al. [60] reported an amplitude modulation study in which conditioning was performed by applying electrical stimuli in a shuttle box. Although not exactly the same conditioning, the behavior between sound stimuli was defined as inter-trial crossings (ITC), which are a measure of activity mainly due to stress in animals. It is important to note that Depner's study was not an exper-iment that examined tinnitus. Kizawa’s method [16] had already been reported as an established method for evaluating tinnitus behavior; thus, we believed that it would be possible to evaluate tinnitus by using this method. However, we cannot rule out the possibility that the FP scores we evaluated are affected by other factors such as stress in addition to active avoidance due to tinnitus.”(in lines 424-435)
Specific comments
Abstract
Point 4: 15 is “Kampo” needed here?
Response 4: Thank you for your suggestion. We did not think it was necessary to include Kampo in this context. Since we wanted to express it more clearly, we changed the expression to “Goshajinkigan (GJG) is one of the formulations of Japanese traditional herbal medicine and is prescribed for the palliative treatment of patients with tinnitus”.(in lines 16-17)
Point 5: 19 change “behavior indicative” to “”behavioral correlates”
Response 5: Thank you for your suggestion. I am sorry, but we changed the relevant parts by the opinions of other reviewers. If the word "behavior indicative" appears in other contexts, we will change it.
Point 6: 20 for three consecutive days
Response 6: Thank you for your suggestion. The word "consecutive" was missing, we added it.
Point 7: 21 after each salicylate injection
Response 7: Thank you for your suggestion. The word "each" was missing, we added it.
Introduction
Point 8: 37 “…various behavioral tests [6]” you can also add Turner et al (2006) here and mention that this is not a conditioned response test but a reflex based test.
Response 8: Thank you for your suggestion. Inserting [6] in this position misleads the reader into thinking that Turner’s experiment is a conditioned response, so we have added the following and inserted [6]. “…various behavioral tests, namely unconditioned gap detection startle reflex procedure [6],”.
Point 9: 48 For the patients it would be nice to know, how fast and how long the effect was, that has been observed. This is also important for the time you choose for your experiments.
Response 9: Thank you for your question. Unfortunately, no experiments have investigated how long the effect lasts in humans. Since the components of GJG have been shown to be maximal in rat blood at 0.5-1.0 hr (Kono, et al. 2015), we examined the behavior and c-Fos expression at 1 hr after GJG administration.
Methods
Point 10: 69 For the time of day, can you provide information when this was relative to light off? Was this also the case during test conditions?
Response 10: Thank you for your suggestion. The laboratory will be turned off at 8:00 PM. We added “(lights off at 8:00 PM)” in the paragraph "2.1. Animals". All tests were performed after lights were off, but conditioning was sometimes performed during the daytime.
Point 11: 70ff As mentioned above, the whole section 2.2 should be rewritten and Fig. 1 should be also revised. Nevertheless, I’ll provide the comments to important information that is missing in the section.
Response 11: Thank you for your suggestion. We were rewritten the whole section 2.2 and revised Figure 1.
Point 12: 75 the standard term for the “conditioning box” is shuttle box
Response 12: Thank you for informing us of the correct terminology. We changed the word " conditioning box " to " shuttle box ".
Point 13: 80 define pure tone frequencies
Response 13: Thank you for your suggestion. We added " with a frequency of 12, 16, or 20 kHz " after the “pure tone” description.
Point 14: 81f the footshock started 1s after the end of the sound stimulus
Response 14: Thank you for your suggestion. The word " the end of " was missing, we added it.
Point 15: 85 change “section” to “session”
Response 15: Thank you for your suggestion. We changed the word " section " to " session".
Point 16: 86 When was the response of the animal in the Shuttle box counted as correct? Please explain this here and not in the next paragraph. I thought that if you have a inter trial time (ITT) of 1 min and a response time of just 1 sec the animal has hardly any chance to avoid the shock completely. You see, that I completely misunderstood the paradigm initially!
Response 16: Thank you for your suggestion. We are sorry for the lack of clarity.
If the rat moved to the next chamber within 6 seconds (5 seconds during the sound presentation plus 1 second before the footshock), we counted it as a correct response. The sentence pointed out was changed as follows: “The ratio of correct avoidance movements in response to sound stimuli within 6 s (5 s during the sound presentation plus 1 s before the electrical footshock) was designated as the active avoidance (AA) score.
We removed duplicate mentions in the “Behavioral Tests” paragraph. 
Point 17: 89 FP: this is only counted in the 1 min ITTs. You write it in the figure caption but not here
Response 17: Thank you for your suggestion. We are sorry for the lack of clarity.
We added, “The number of movements to the opposite side without sound stimulation during the intertrial interval was defined as the false positive (FP) score.” (in lines 91-93)
Point 18: 91 80%: how long did the training take. Can you provide learning curves?
Response 18: Thank you for your suggestion. We created learning curves in supplementary Figure 1.
Point 19: Figure 1: B and C can be united; the control group (saline) is missing completely and as I understood all animals received something (water/GJG) orally.
Response 19: Thank you for your suggestion. We united Figures 1B and C.
In goshajinkigan (GJG) administration protocol (B2), the control group (saline + water) was set. I am sorry for the confusing writing style. We added, “For the control group, animals received daily i.p. saline injections and following oral administra-tion of DW..” In Figure 1 caption.
Point 20: 138 Explain, that these are different animals.
Response 20: Thank you for your suggestion. We added “Separate from the behavioral tests,“ .(line in 139)
Point 21: 173ff As mentioned above, change the statistics accordingly.
Response 21: Thank you for your suggestion. We changed “the mean ± SEM” to “the median [interquartile range (IQR) 25-75]”. We removed the statement that we used parametric tests.
Results
Point 22: Table 1: also here use median and 25-75% values
Response 22: Thank you for your suggestion. We changed “the mean ± SEM” to “the median [interquartile range (IQR) 25-75]”.
Point 23: 191ff: As mentioned above, please change the statistics where necessary.
Response 23: Thank you for your suggestion. We turned the results of parametric tests into those of nonparametric tests. We changed “the mean ± SEM” to “the median [interquartile range (IQR) 25-75]”.
Point 24: 196 Why do you start with Fig. 2C? Is it possible to rearrange the order of the figure or the order in the text?
Response 24: Thank you for your suggestion. We explained it in the earlier order to show that the administration of salicylate could increase FP scores. However, as you pointed out, it was difficult to understand, so we rearranged the order in the text.
Point 25: 199f “The FP scores…” you just said that they were not different. Please clarify.
Response 25: Thank you for your suggestion. We put “At a stimulus of 16-kHz frequency, “ in front of the sentence “comparison of the FP scores on day 3 among groups exposed to different sound inten-sities revealed that the scores were higher at 40 dB and 60 dB than at 80 dB” for clarity. (in line 193)
Point 26: 203 Which FP scores are investigated?
Response 26: Thank you for your suggestion. I am sorry for being so confusing.
We changed “When the FP scores were compared for different frequencies of the sound stimulus, no significant …” to “At a stimulus of an intensity of 60 dB SPL, comparison of the FP scores on day 3 among groups exposed to different sound frequencies revealed that no significant …”. (in lines 200-202)
Point 27: Figure 2: separate the Figure into two, as A-C investigate something completely different than D-F. In line 214 you show the remaining animal groups for the first time!
Response 27: Thank you for your suggestion. We divided Figure 2 into A-B and C-F. C-F was set as new in Figure 3. The number of animals had been only mentioned in Figure 2 caption. We also wrote the number of animals in the Methods. We were sorry for not mentioning it in the Method but only in the Figure caption.
Point 28: 233 again, which FP scores are investigated?
Response 28: Thank you for your suggestion. We put “on day 3” after “the FP scores” for clarity.
Point 29: 238 No mentioning of Figure 2E in the results?
Response 29: Thank you for your suggestion. Figure 2E was illustrated at the same time as Figure 2F, so we had changed “…(Figure 2F)” to “…(Figure 2E, 2F)”. Since we renumbered our graphs, we changed “…(Figure 3D)” to “…(Figure 3C, 3D)”.
Point 30: 249 Why do you use these tests? Why not a Kruskal-Wallis ANOVA with post-hoc tests? If you use MW U tests, did you correct for multiple testing (e.g. Bonferroni)?
Response 30: Thank you for your suggestion. Regarding the graph comparing the number of c-Fos positive cells between three groups, we believe that they are not comparable in terms of “the main effect”.
A comparison of Saline + DW and SS + DW can examine the difference in effectiveness between Saline and SS. A comparison of SS + DW and SS + GJG can examine the difference in effectiveness between DW and GJG. However, comparisons between Saline + DW and SS + GJG cannot be made because there is nothing to compare.
This method was taught by Dr. Inoue (written in Acknowledgements) who is an expert in statistics.
Point 31: 260 why a t test? Anyway, do not use parametric tests.
Response 31: Thank you for your suggestion. We retested and rewrote with nonparametric tests.
Point 32: Figure 3 and 4: Remove the statistics from the figure captions. Put them in separate tables, use median and 25-75%.
Response 32: Thank you for your suggestion. We removed the statistics from the figure captions of Figures 3 and 4 (new Figures 4 and 5), and created Tables 2 and 3. We used median and 25-75% for all Figures.
Discussion
Point 33: 316f 40 vs 60 vs 80 dB: As mentioned above, this could indicate, that the animals did not react to tinnitus but were more stressed, when the task was more difficult. The FP you are counting are usually seen as ITCs which indicate the stress level of the animals. Please discuss this.
Response 33: Thank you for your suggestion. We used an established method for assessing tinnitus. We examined behaviors that increased when salicylate-induced tinnitus was produced. We believe that this is similar to the gap detection test.
On the other hand, the ICT is an indicator that increases during stress, but it is not evaluated by administering salicylate and is not considered the same experiment, although a similar experimental setup is used. We added several sentences to the Discussion.
Point 34: 318 12-kHz, 16-kHz, and 20-kHz stimuli: in the context of stress (but not tinnitus) this makes sense as the frequencies all are well in the area of best hearing. Nevertheless, tinnitus and stress might be closely related. But is needs to be discussed.
Response 34: Thank you for your suggestion. Since the relationship between our experiment and stress cannot be completely ruled out, we have added our ideas in the last paragraph of the discussion.
Point 35: 325 hearing loss: at least not in the range that affects the detection. See Lanaia et al. (2021)
Response 35: Thank you for your suggestion. The paper by Lanaia et al. was very helpful in understanding salicylate-induced tinnitus and hearing loss.
Point 36: 327 functional damage to endocochlear organs: did you look for bleedings? A high dose of salicylate might induce them. In gerbils, doses of "only" 300 mg/kg sometimes lead to bleedings in the cochlea.
Response 36: Thank you for your suggestion. We did not look for bleedings. If your suggestion that a high dose of salicylate might induce bleedings is true, our textual representation seemed too strong. We pulled the word “any” out of the sentence “salicylate did not induce any obvious functional damage to endocochlear organs”.(in line 346)
Point 37: 331 patients: is it known, if GJG has stress/anxiety releasing effects in humans? If FP is - at least partially - a stress indicator, it could explain parts of the effect. Else, it would point to a more tinnitus related effect.
Response 37: Thank you for your question. Clinically, GJG is not administered for stress/anxiety relief. We believe FP is an indicator of established tinnitus behavior.
Point 38: 339 1 g/kg GJG: have these concentrations ever been tested in humans? They might have other/additional effects than the low doses (in animals factor 10 is quite high compared to humans, most of the time, factor 3 (your low dose) is more common)
Response 38: Thank you for your suggestion. These concentrations have never been tested in humans.
Animal studies using GJG often use a dose of 1 g/kg.
Mizuno, K. et al (2014) used doses of 0.3 g/kg and 1 g/kg when analyzing rat models, while PCR studies compared GJG 1.0 g/kg to a control group.
Kono, T. et al (2015) used 1 g/kg GJG for rats.
Hagihara, K., Nunomura, K., and Lin, B. et al (2022) used 1 g/kg GJG for mice.
This amount is generally high for humans and animals, but we used this amount because we judged it to be a common amount in GJG's experiments.
Point 39: 353 maybe mention what the exact differences of the two mentioned studies compared to your study are.
Response 39: Thank you for your suggestion. Wu et al. administered 250 mg/kg salicylate for 5 days, and Santos et al. administered 300 mg/kg salicylate for 3 days. In addition to these conditions, the time from SS administration to fixation was added as it differed significantly. We added “…and time from SS administration to fixation (250mg/kg salicylate for 5 days fixation after 8 h [19], 300mg/kg salicylate for 3 days fixation after 3 h [21]).” (in lines 376-378)
Point 40: 362f “central auditory system”: here you could discuss the effects found in Lanaia et al. 2021
Response 40: Thank you for your suggestion. The discussion on salicylate-induced tinnitus in Lanaia et al. 2021 was really interesting.
We added “This agrees with the salicylate-induced tinnitus model reported by Lanaia et al. [38]. They adopted the behavioral assessment of tinnitus using somatic resonance where the DC played an important role. Their results indicate that the DC is not involved in sa-licylate-induced tinnitus, but is involved in trauma-induced tinnitus.” (in lines 369-372)
Point 41: 382 The statement in your concluding sentence was already known. You showed that your model might be capable of being used for the investigation of the described effect.
Response 41: Thank you for your suggestion. We created a new paradigm of "conclusions" and noted a summary.

Round 2
Reviewer 1 Report
The authors have provided a detailed point-by-point answer to all the questions raised, with appropiate arguments where needed and have done the modifications in their Ms according to the established scientific discussion.
The final result is satisfactory and no further rounds are needed.
Reviewer 2 Report
Thank you for putting so much efford in rewriting large part of the manuscript. I see no more obstacles for publication.